# Screen Time and Its Health Consequences in Children and Adolescents

**DOI:** 10.3390/children10101665

**Published:** 2023-10-08

**Authors:** Nikos Priftis, Demosthenes Panagiotakos

**Affiliations:** 1School of Medicine, National and Kapodistrian University of Athens, 115 27 Athens, Greece; nk.priftis84@gmail.com; 2School of Health Sciences and Education, Harokopio University, 176 71 Athens, Greece

**Keywords:** screen time, digital media, children, adolescents, review

## Abstract

Nowadays, children and adolescents are exposed to digital media (DM) from an early age. Therefore, specific guidelines have been published by the World Health Organization, whose aim is to limit daily screen time (ST) viewing. However, during the COVID-19 pandemic, a rise in DM use, and consequently ST viewing, was observed. More and more aspects of modern life are thought to be affected by excessive ST viewing. Accordingly, the aim of this review is to document the health effects of excessive ST viewing on children and adolescents. A narrative review was performed in searchable databases. In total, 43 original articles were considered. Excessive ST viewing was correlated with increased risk for obesity and other cardiometabolic risk factors, mental health, unhealthy dietary habits and eating disorders, and problems in development and child–parent relationships. Sleep, physical activity, eyesight, headaches, and the musculoskeletal system were negatively affected as well. However, the effect of ST was weighted by the type of media used and the way types of media were used. Other confounding factors were reported. There is evidence to suggest a negative correlation between excessive ST and youth health exists. Nevertheless, more research is needed if this correlation is to be established.

## 1. Introduction

Children and adolescents grow up in a society where modern technology has become a crucial part of their daily lives. As a result, they are exposed to electronic devices from an early age and consequently to screen time (ST) viewing. According to the World Health Organisation (WHO), ST is defined as: “Time spent passively watching screen-based entertainment (TV, computer, mobile devices). Does not include active screen-based games where physical activity or movement is required.”. Additionally, according to WHO’s guidelines, children under the age of 24 months should not be exposed to screens, whereas children under the age of 5 years should only be exposed to screens for a maximum of 1 h daily [1].

Recently, the interest in studying ST has surged due to the COVID-19 pandemic. According to Sheri Madigan et al. [2], there has been a 52% increase in ST amongst children and adolescents during the COVID-19 pandemic. Concerns over the risks of ST are not something new. Since 1985, Dietz WH et al. published one of the first studies associating excessive ST with obesity [3]. Moreover, apart from health consequences, the growth of digital media (DM) is believed to have a socioeconomic impact as well. Although there is evidence to suggest that excessive ST is harmful to youth, in a recent study, more than 80% of the participating parents reported that they believe that ST increases their children’s creativity and imagination [4]. However, in another study, it was reported that 90% of the participating households had ST rules [5].

Several studies have already revealed that excessive ST can lead to the development of certain cardiometabolic disorders, mainly obesity and high blood pressure, as well as sleep disorders, chronic neck and back problems, depression, and anxiety. Moreover, excessive ST has been associated with low performance at school in children. However, the health consequences of ST among children and adolescents have not yet been well understood and appreciated. There is a controversy over studies’ findings that balance between benefits and harm. Thus, the aim of this review is to present the health effects of ST viewing among children and adolescents.

## 2. Materials and Methods

### 2.1. Literature Search and Study Selection

A comprehensive review was performed in searchable databases, i.e., Medline (through PubMed), Scopus, CINAHL, and Communication & Mass Media Complete, using common keywords and Medical Subject Headings (MeSH), and the logical terms AND/OR, i.e., [“screentime” OR “screen time” OR “social media” OR “digital media” OR ”video game” OR ”video games” OR ”video gaming” OR “smartphone” OR “smartphones” OR “mobile phone” OR “mobile phones”], AND [“teen” OR “teens” OR “toddler” OR “toddlers” OR “children” OR “youth” OR “teenager” OR “teenagers” OR “adolescent” OR “adolescents” OR “adolescence”], AND [“health” OR “obesity” OR “cardiometabolic” OR “cardiovascular” OR “adiposity” OR ”blood pressure” OR ”metabolic” OR “mental” OR “emotion” OR “emotions” OR ”emotional” OR “depression” OR “depressive” OR “anxiety” OR “inattention” OR “attention” OR “ADHD” OR “aggression” OR ”suicide” OR ”suicidal” OR “diet” OR “dietary” OR “eating” OR “development” OR “developmental” OR “behavior” OR “behaviour” OR “behavioral” OR “behavioural” OR “language” OR “autism” OR “academic achievements” OR “parent” OR “parents” OR “parental” OR ”sleep” OR ”sleeping” OR “physical activity” OR “playing outside” OR “outdoor” OR “eye” OR “eyesight” OR “myopia” OR ”headache” OR ”headaches” OR “migraine” OR “musculoskeletal” OR “scoliosis” OR “arthritis” OR “pain”]. Original research studies in children and adolescents were reviewed; the reference lists from previous reviews and meta-analyses were also evaluated to allocate studies that were missed during the literature search. Meta-analyses and reviews were considered in order to compare the findings of the present study with previous research.

It was decided that only studies published in the period of 2017–2023 would be considered. In 2020, the WHO declared the COVID-19 pandemic as a public health emergency of international concern and ended the declaration in 2023. Therefore, the duration of the pandemic as a public health emergency was 3 years. Consequently, selecting studies from 2017 onwards would cover an equal number of years before and during the pandemic.

### 2.2. Inclusion and Exclusion Criteria

Inclusion criteria: (i) English language; (ii) published from 1st of January 2017 until July 2023; (iii) target population aged from infancy up to 20 yo (so that no effects would be missed); (iv) evaluation of sedentary ST; (v) assessment of distinct outcomes of the effect of ST on health.

Exclusion criteria: (i) evaluation of non-sedentary ST; (ii) assessment of parental ST only; (iii) target population aged over 20 yo; (iv) inadequate data on ST measurements or ST’s effect on health; (v) sample size less than N = 100 for cross-sectional studies and less than N = 50 for longitudinal studies.

### 2.3. Data Extraction

One author (N.P.) extracted data from the selected studies using the following: publication year, study design, sample size and age, ST evaluation, and the effect of ST on health.

## 3. Results

### 3.1. Selection of Studies

A literature search was conducted in PubMed, Scopus, CINAHL, and Communication & Mass Media Complete. With the previously mentioned search terms, 3357 results were found. After the duplications were removed, inclusion and exclusion criteria were applied, the titles and abstracts were considered, and the selection of studies was finalised. In total, 43 original studies were critically reviewed in this work. They consisted of 29 cross-sectional studies, 10 longitudinal studies (the study of Pedersen J et al. [6] has both a cross-sectional and a longitudinal aspect), 3 case-control studies, and 2 random crossover clinical trials. A PRISMA flow diagram is presented in Figure 1, depicting the study selection process.

Moreover, data from 10 meta-analyses, 10 reviews, 1 position statement, and 1 meta-synthesis were considered to more fully discuss the findings of the present paper. 

### 3.2. Obesity

Obesity is one of the first health consequences of excessive ST that has been investigated [3]. Nowadays, it is widely accepted that excessive ST correlates with childhood obesity [7,8,9,10,11,12,13]. According to a meta-analysis by Fang K et al., more than 2 h of daily ST is positively associated with obesity. The association was greater for TV and computer ST viewing [10]. This association was found to be dose-dependent by Zhang G et al. Each added hour of daily ST increased the chance of obesity by 13% [11]. The type of ST also plays an important role, as social media (SM) use has been described to be the most important risk factor (RF) for obesity in primary and secondary school children [12], while for video games (VGs), no such association could be found [13]. Mineshita Y et al. made a distinction between the timing of ST (before bedtime) and the duration of ST, both of which were linked with obesity [14]. The correlation between excessive ST and obesity is stronger amongst older children and adolescents. That is to say, it can still be found in younger children, as proven in a longitudinal study by Schwarzfischer P et al. [15]. It should be noted that ST not only affects the weight index but also affects adiposity levels [16]. The mechanisms with which ST affects body weight are yet to be confirmed. However, ST might influence the satiation signals and the habitual control of food intake, leading to higher caloric intake [17]. Ad viewing can also increase caloric intake and unhealthy food consumption [18], heightening the risk for obesity. It should be noted that ST is clearly linked with lower physical activity (PA) levels, which in turn are linked with obesity [19]. Refer to Section 3.5. “Dietary habits, Eating disorders” for further relative information.

### 3.3. Other Cardiometabolic Risk Factors

Excessive ST is thought to affect the cardiovascular system according to various researchers [7,9,20,21,22], and a dose-dependent negative association has been documented after exceeding a 2 h threshold of daily ST [20]. Cureau FV et al. documented that excessive ST affects cardiometabolic risk only in overweight children [21]. Cardiovascular disease (CVD) studies have mainly focused on cardiovascular fitness (CVF), blood pressure (BP), insulin resistance (IR), and cholesterol levels. A large-scale European cohort has associated all these factors with excessive ST across childhood [22]. Additionally, more than 3 h of ST may lead to IR and elevated adiposity markers in children aged 9–10 yo, predisposing them to type 2 diabetes, even though no links between ST and blood glucose were found [16]. A Swedish study of 13 yo children found that CVF is negatively associated with more than 5 h of weekday ST for boys and more than 3–4 h of weekday ST for girls, while participation in organised sports was positively associated [23]. Moreover, ST on both weekdays and weekends was described as an RF for cardiorespiratory endurance of children and adolescents [24]. Interestingly, a cohort by Pedersen J et al., in contrast to other studies, reported no correlation between BP in toddlers and ST duration. However, a cross-sectional analysis of the same study linked ST before sleep and elevated BP [6]. At the same time, Vanderloo LM et al. could only identify a very subtle effect of ST on CVD. Specifically, excessive ST caused a moderate, dose-dependent reduction in HDL [25]. Refer to Section 3.2. “Obesity” for further relative information.

### 3.4. Mental Health

It has been well established that excessive ST affects the mental health of youth [7,8,9,26]. Recently, pandemic-related stressors decreased PA and increased ST, both associated with worse mental outcomes [26]. Mental problems can be categorised as internalised (depression, anxiety) or externalised (attention-deficit/hyperactivity disorder (ADHD), behavioural problems). In an interesting cohort, Marin-Dragu S et al. support that active use of smartphones could be objectively measured by the number of times one unlocks their smartphone, as “number of unlocks”, while passive use by the total duration of ST, as “duration of ST”. It was found that the number of unlocks is a protective factor (PF) against externalising symptoms, and the duration of ST is an RF for both internalising and externalising symptoms. Number of unlocks’ protective effect is reduced as the duration of ST increases [27].

#### 3.4.1. Internalising Symptoms

The main forms of internalising symptoms are depression and anxiety. A meta-analysis by Liu M et al. on teenagers found a dose-dependent correlation between SΤ and depression. Further study showed that girls had an almost 1.5 times higher probability of displaying depressive behaviours compared to boys for every added hour of ST. Contrary to other findings, which suggest that SM mainly affects younger children and not adolescents, teenagers were also negatively affected by SM [28]. However, difficulties in the analysis of age as a confounding factor were described. Regarding the difference in the prevalence of depression between girls and boys, Hökby S et al. had some indicative findings. They focused on two types of coping mechanisms: (i) Problem-focused engagement coping (PFE), including problem solving and cognition reconstruction, and (ii) Emotion-focused engagement coping (EFE), including emotional expression and social contact. Excessive ST led to depression more frequently when PFE was weaker, unlike EFE, where no interactions were found. This means that actively trying to overcome one’s problems is effective while trying to externalise them is not. Boys engaged more in PFE compared to girls. Therefore, the difference in the prevalence of depressive symptoms could be explained [29]. Additionally, a scoping review by Hilty DM et al. states that more than 3 h of daily ST results in a higher chance of depression and anxiety in adolescents and young adults [30]. A longitudinal study reported that TV, computer use, and VGs had a partially bidirectional relationship with emotional disorders. More specifically, more than 4 h daily of TV ST led to panic disorder symptoms, and more than 4 h daily of computer or VG ST led to anxiety and social phobia symptoms, even though PA prevented anxiety symptoms. Reversibly, no association was found between pre-existing panic disorder symptoms and ST, while pre-existing depression or anxiety symptoms led to more computer and VG usage [31].

#### 3.4.2. Externalising Symptoms

The main forms of externalising symptoms are ADHD and behavioural problems such as aggression. Michal Kahn et al. reported that more than 2 h of daily ST correlates with externalising problems in preschool children, though this was only true in children who lacked sleep [32]. A cohort study by Tamana SK et al. documented that more than 2 h of ST when children were aged 3 or 5 yo led to more externalising problems at the age of 5. Further analysis of these results showed that clinically evident inattention is associated with excessive ST, unlike aggression [33]. The same cohort revealed that more than 2 h of weekly exercise was protective against externalising problems, and it is a well-established fact that excessive ST negatively affects PA levels [8]. Thus, excessive ST can affect externalising behaviours indirectly. Young adolescents exposed to explicit sexual media are at a higher risk for embracing high-risk sexual behaviours (early sexual debut, unsafe sex, higher number of sexual partners) in later adolescence. The effects were stronger when more media types were utilised [34].

#### 3.4.3. Suicidal Behaviours

Suicidal behaviours cannot be exclusively classified as internalising or externalising behaviours, so they are studied separately. DM has introduced a new form of bullying—cyberbullying. Cyberbullying has been linked with suicidal ideation (SI), suicidal behaviour (SB), suicide attempts (SA), and self-harm (SH) in youth aged less than 25 yo. More specifically, victims have a 2.5 times higher chance of presenting SI and SB, SA, and inflicting SH, while perpetrators have a higher chance of presenting SI and SB. A school-aged-children-only sub-analysis established equivalent results in young adolescents. Moreover, cybervictimisation has a greater impact than traditional victimisation, but when both are present, traditional victimisation lessens cybervictimisation’s effect [35]. It should be noted that the heterogeneity of the studies was significant, and as Stiglic N. et al. comment, there is a lack of studies on the specific subject [9].

#### 3.4.4. Healthcare Setting

Interestingly, there have been thoughts of introducing VGs in healthcare settings. In a review article, María Rodrigo-Yanguas et al. support that specialised and educational VGs can be used to alleviate the symptoms of ADHD in children [36]. The same could be said for in-hospital stressors, as Sajeev MF et al. concluded that distracting VGs are associated with less in-hospital anxiety and pain in paediatric populations. However, caregivers were not affected [37].

### 3.5. Dietary Habits and Eating Disorders

Excessive ST may also affect the eating behaviours of youth. An observational study of Brazilian adolescents found that both more than 2 h of daily ST and eating in front of a TV more than doubled the risk of high ultra-processed foods (UPF) consumption [38], while excessive ST was found to prompt Greek youth to eat a less healthy diet and skip breakfast [39]. Moreover, excessive ST on weekdays amongst Swedish adolescents was linked with irregular breakfast patterns, unlike excessive ST on weekends [40]. 

In addition to the dietary influences, ST, especially SM, is linked with eating disorders (EDs) [8,41]. Wilksch SM et al. investigated that relationship in children aged 12–13 yo. The following had to be fulfilled for an ED to be diagnosed: (i) poor body image and (ii) disordered eating behaviours. Girls with two or more SM accounts and boys with three or more SM accounts had higher chances of fulfilling diagnostic criteria for EDs, while a linear association was found between EDs and the time girls spend on Instagram. Interestingly, boys were more at risk for having a poor body image when compared to girls [42]. An Australian study reported that adjusting gender by SM behaviour nullified its effect as an RF for EDs. More specifically, avoidance of posting selfies, photo investment, photo manipulation, and investment in others’ selfies were all correlated with increased risk for EDs. Therefore, screening for these types of behaviours could help in ED prevention [43].

### 3.6. Development

The effects of ST on the development of children and adolescents have been identified by many research groups [6,9,44]. A team from Canada found that excessive ST has a negative impact on aspects of development [45], while other research findings [46] suggest that ST in children younger than 2 yo is associated with developmental delays, supporting the WHO guidelines. Excessive ST in children aged 29 mo hinders their kindergarten acclimatisation since it negatively affects vocabulary, number knowledge, classroom engagement, and locomotion and worsens victimisation, as found by a Canadian population study [47]. Likewise, more than 1 h of ST in children aged 32 mo led to communication and daily living skills deficiencies. Outdoor play was found to act as a mediator in the relation of ST with daily living skills [48]. In older children, ST hinders their academic performance and in-class attention. Longer ST duration and ST before bedtime were both identified as separate risk factors. Interestingly, longer ST duration, when compared to ST before bedtime, had a greater influence on academic performance, possibly due to study displacement by screen use [13]. ST also influences physical strength [49] and overall physical condition [39], cardiorespiratory endurance, and overall health [24], which are all aspects of healthy physical development in school-aged children. A study from Taiwan also revealed that more than 2 h of daily ST affects the motor development of toddlers [50]. Additionally, more than 2 h of ST was linked with deficient language skills, and more than 4–5 h was linked with deficient personal and interpersonal skills and play. The effect was reported to be dose-dependent [4]. It is worth noting that excessive ST has been recognised as a possible RF for chronic conditions. Heffler KF et al. documented that while an early onset of ST viewing does not affect the risk for autism spectrum disorder (ASD), it increases the risk for ASD-like symptoms [51]. Furthermore, recently, the term “digital dementia” has been proposed by researchers who consider excessive ST from an early age and dementia to be correlated [52].

It should be highlighted that the effects of excessive ST are not rigid and may differ by type of screen use. Educational shows, co-viewing with a parent, and postponed regular use of media all promote language skills. In contrast, longer duration of ST and background ST both impede language skills [46]. Additionally, TV viewing has a greater impact on physical strength than VG [49], possibly because TV viewing is a more passive activity, and weekday ST has a stronger link with worsened physical attributes than weekend ST [24]. On the other hand, smart-device usage was found to be positively associated with fine motor movements in children between 24 and 35 months of age, possibly because of the index finger scrolling. There was no such association in older children, meaning that the benefit could wear off over time [53]. Moreover, video chatting seems to be appropriate for learning and social bonding in children under the age of 2 and could be used for educational purposes [54]. Refer to Section 3.4.2. “Externalising symptoms” for further relative information.

### 3.7. Child–Parent Relationship

Parents shape their children’s ST habits. When parents view TV, browse the internet, or use SM for more than 2 h, their children are at higher risk for excessive ST. The same is true if children watch TV alone, without a grown-up supervisor [4]. Moreover, it was found that non-parent care providers granted permission for more ST [50]. Interestingly, the duration of parents’ work did not affect their children’s ST [4]. It should also be noted that a negative child–parent relationship was linked with problematic internet use. This finding was moderated by the type of child–parent relationship, age, and country [55]. A review article by Chong SC et al. studied what parents thought about their children’s ST. Parents used DM as (i) a distraction for their children when they needed a break, (ii) an educational tool for their children (especially true for the interactive use of computers), and (iii) a reward or penalty system [44]. The last one has been documented as an RF for parent–child conflict [56]. According to the aforementioned review, parents had diverse opinions about the perception of their children’s ST. They considered it as something essential and unavoidable or as something unsettling and possibly unnatural [44].

### 3.8. Sleep and Physical Activity

Kahn M et al. used actigraphy to study sleep patterns and ST. Their findings suggest that excessive ST leads to poorer quality and shorter duration of sleep and later sleep onset [31]. Moreover, a questionnaire-based study found that ST influences sleep in infants and toddlers. No ST was linked with more total sleep time compared to any duration of ST in children aged 0–2 yo [57]. Toddlers also seem to be affected by excessive ST. More than 1 h daily of TV led to increased sleep problems. No such association could be established with computer or tablet use; thus, more research is needed [58]. Research in young adults suggested that REM sleep duration is affected by ST, too [59,60]. The mechanisms with which ST interferes with sleep have been mainly studied in the adult population. The three major hypotheses are: (i) screen light directly suppresses melatonin levels [59], (ii) DM causes mental arousal [60], and (iii) ST displaces sleep time [61].

Excessive ST was associated with lower PA intensity levels [24] and less playing outside [5,62] in children and adolescents. Moreover, in a longitudinal study, excessive ST at 2 years of age led to less time playing outside 8 months later [48]. Unlike ST duration, no interactions have been found between ST before bed and PA [13]. Interestingly, according to a cross-sectional study by Dahlgren et al., objectively measured smartphone use does not affect PA levels in teenagers [63]. Furthermore, it should be highlighted that adequate PA can mitigate many of the negative effects of excessive ST [19,20,48,64]. Refer to Section 3.6. “Development” for further relative information.

### 3.9. Eyesight and Headaches

A meta-analysis by Foreman J et al. supports that there is evidence to suggest that a correlation between myopia and ST in smart devices does exist. However, the literature is insufficient, and no conclusion can be drawn [65]. Another notable eye-related clinical entity induced by excessive ST is “digital eye strain” (DES). According to Kaur K et al., DES’s symptoms can be classified as follows: (i) related to reduced blink and tear formation (dry eyes, headaches, eye discomfort), (ii) related to eye focus and synchronised movement (blurry vision, distance refocus), or (iii) musculoskeletal (headaches, neck pain) [66]. An explanation for why excessive ST leads to dry eyes and eye discomfort could be that high-concentration tasks (such as concentrating on a screen) reduce blinking rate and tear break-up time [67], causing dryness and refraction problems. Interestingly, eye pain has been identified as a PF against excessive ST [62].

ST has also been linked with migraine headaches [68,69,70]. Montagni I et al. have proposed two possible mechanisms: (i) screen light’s wavelength per se may provoke a migraine attack, or (ii) ST lowers the threshold that triggers a migraine attack due to other causes [70]. No correlation could be found for other types of headaches [69,70].

### 3.10. Musculoskeletal System

A meta-analysis by Baradaran Mahdavi S et al. proposed that excessive ST’s effect on neck pain is borderline insignificant. Nonetheless, a significant association was established by excluding one study [71]. Furthermore, mobile phone use is documented as an RF for neck [71] and back [64] pain in youth. Computer use’s relationship with back pain was not as strong [64]. In another study, more than 2 h of ST on weekdays was linked with adolescent idiopathic scoliosis (AIS). The same was true for low dairy product consumption [72], which, in turn, has been linked with excessive ST [37,39]. Additionally, Tremblay MS et al. suggested that less than 2 h of daily ST is associated with better bone health [20].

## 4. Discussion

The findings of the present study mostly agree with those of previous research when compared to previous systematic reviews [8,9,20,30,35,36,41,66] and meta-analyses [10,11,13,28,46,55]. Trends in DM emerge and fade away regularly, and consequently, the way people use DM evolves. As a result, investigations should be continuous since newer data might differ from the already established ones. In this review, the latest studies are considered, and the chance of omitting data is minimised. Moreover, looking at the research conducted before and during the pandemic, it is evident that the pandemic has prompted research on the effects of ST on health, and it has made scientists more aware of its impact.

Interestingly, different types of ST have different outcomes on health. According to a study by Zhu R et al., only TV viewing is associated with worse sleep outcomes in preschool children [58]. However, when compared to VGs, TV viewing has a lesser impact on mental health [31]. Smartphone use, when studied independently, is not correlated with reduced PA [63], unlike TV viewing [49] and nonspecific ST viewing [13,22,39]. It should also be noted that passive ST was linked with worse developmental outcomes when compared to interactive ST [51,54].

Furthermore, the age of children was identified as a modifying factor. A greater influence on BMI [12] and CVF [24] was observed in adolescents. Additionally, a study showed that as children grow older, they tend to use DM more. If this increase is more significant than what would be considered normal, physical fitness levels can be negatively affected. The timing of ST was found to moderate the effects of ST. More specifically, excessive ST on weekends was associated with increased BMI and decreased CVF [23], while excessive ST on weekdays was associated with breakfast skipping [40]. Also, ST before bedtime was linked with dry eyes [13] and increased BP in young children [6]. Lastly, gender was observed to mediate the effects of ST. Excessive use of DM in girls was more likely to negatively affect CVF [24] and mental health [29] and increase EDs [42,43]. However, when stratified for PFE [29], and avoidance of posting selfies, photo investment, photo manipulation, and investment in others’ selfies [43], gender’s significance was nullified. Therefore, screening for such behaviours could prove useful for prevention.

All the studies and meta-analyses are presented in Table 1. Further discussion points are addressed in the Section 3.

This research, as with any similar study, has strengths and limitations. As the main strengths, the following could be considered: In the last 10 years, even if the major digital devices (i.e., TVs, computers, smartphones, gaming consoles, and tablets) have evolved, their function has remained fundamentally the same. Thus, the usage of DM could be conventionally considered unchanged. Additionally, due to the small number of authors, subjective opinions had a minimal impact on the results of this study. Mechanisms by which excessive ST affects health were investigated as well.

As the main limitations, the following could be considered: The ways DM is utilised have remained unchanged only over the last few years. Older data that was retrospectively collected in some studies might not accurately depict DM’s health effects on children and adolescents today. Furthermore, many confounding factors that moderate the relationship between ST viewing and the health of children and adolescents were not considered in all of the studies. The influence that cultural and socioeconomic differences between populations may have was not taken into account.

## 5. Conclusions

Excessive ST has many negative health effects caused by a multitude of mechanisms, affecting plenty of systems. Specifically, there is evidence to suggest a negative impact on obesity, development, CVD, mental health, diet, child–parent relationship, sleep, PA, eyesight, headaches, and the musculoskeletal system. However, more research is needed, not only for more accurate relationships to be established but also for confounding factors to be clarified. To prevent the negative effects of excessive ST from appearing, paediatricians and other healthcare professionals should inform parents about its dangers and encourage them to abide by the corresponding WHO guidelines. Moreover, screening for certain behaviours that predispose to negative health effects of ST should be applied and further studied.

## Figures and Tables

**Figure 1 children-10-01665-f001:**
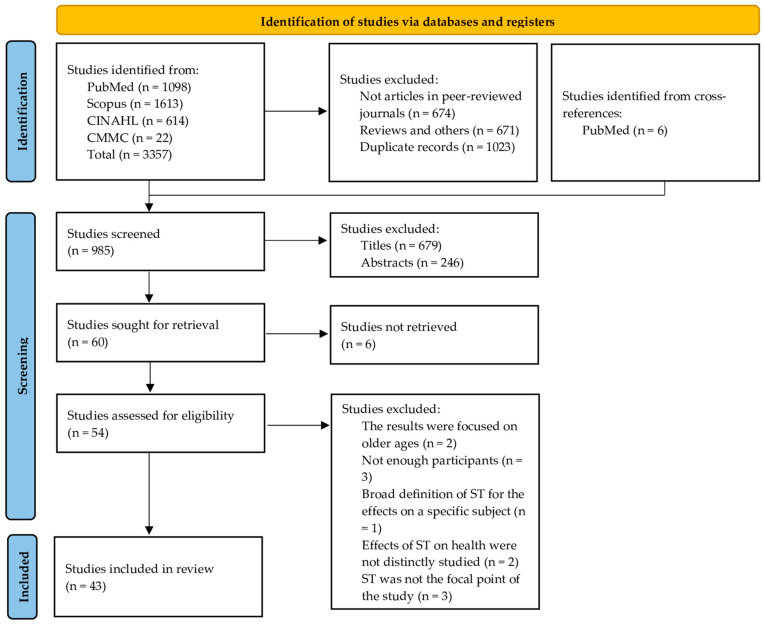
PRISMA flow diagram for study selection process.

**Table 1 children-10-01665-t001:** A concise presentation of the studies and meta-analyses used.

Author	Year	Type	Sample (N)	Main Findings
Dy ABC et al. [4]	2023	Cross-sectional	419	More than 2 h of daily ST was negatively associated with toddler development. Co-viewing was a protective factor against excessive ST, while background media and parents with excessive ST were risk factors.
Hinten AE et al. [5]	2022	Cross-sectional	356	Daily activities of primary school children were studied by analysing diary entries filled out by parents.
Fang K et al. [10]	2019	Meta-analysis	n/a ^1^	More than 2 h of daily ST, TV, or ST and computer use was associated with obesity.
Zhang G et al. [11]	2015	Meta-analysis	n/a ^1^	Excessive ST was positively associated with obesity. For every hour of ST, the risk of obesity increased by 13%.
Khajeheian D et al. [12]	2018	Cross-sectional	1569	Excessive ST was positively associated with obesity. The association was stronger amongst high school students when compared to primary school students.
Haghjoo, P et al. [13]	2023	Meta-analysis	n/a ^1^	Excessive ST was associated with a 20% higher risk for obesity. No dose-dependent effect could be established.
Mineshita Y et al. [14]	2021	Cross-sectional	6334	Longer ST duration was positively associated with obesity and negatively associated with PA and learning ability. ST before bedtime was positively associated with obesity and negatively associated with dry eyes and learning ability.
Schwarzfischer P et al. [15]	2020	Longitudinal	526	More than 1 h of daily ST was positively associated with obesity in children aged 3–6 yo. At the ages of 3–6 yo, 1 h of additional ST led to increased risk for obesity at the age of 6 yo.
Nightingale CM et al. [16]	2017	Cross-sectional	4495	More than 3 h of daily ST was positively associated with adiposity and type 2 diabetes risk factors in children aged 9–10 yo.
Bellissimo N et al. ***** [17]	2007	Case control	14	TV ST affects satiation signals, leading to higher caloric intake during lunchtime in boys.
Russell SJ et al. [18]	2018	Meta-analysis	n/a ^1^	TV viewing and advertisements in VGs were associated with higher caloric intake in children and adolescents.
Cureau FV et al. [21]	2016	Cross-sectional	36,956	Equal or more than 6 h daily of ST was positively associated with cardiometabolic risk in overweight adolescents aged 12–17 yo.
Sina E et al. [22]	2021	Longitudinal	10,359	DM use increases with age. Excessive ST was positively associated with metabolic syndrome and cardiometabolic risk in both sexes. More than normal ST increase was positively associated with metabolic syndrome in both sexes and insulin resistance in boys. PA was a protective mechanism.
Kjellenberg K et al. [23]	2021	Cross-sectional	884	More than 5 h of ST on weekends was negatively associated with cardiovascular fitness levels in boys aged 13 yo; 3–4 h or more of ST on weekends was negatively associated with cardiovascular fitness levels in girls aged 13 yo.
Hardy LL et al. [24]	2018	Cross-sectional	6405	Longer ST was negatively associated with PA attributes; the association was stronger for adolescents, girls, and weekday ST.
Pedersen J et al. [6]	2021	Longitudinal	628	The change in total ST from the age 3 yo to 5 yo was not associated with increased BP or increased BMI measurements.
Pedersen J et al. [6]	2021	Cross-sectional	964	Excessive ST was not associated with increased BP in children aged 5 yo.
Pedersen J et al. [6]	2021	Cross-sectional	963	With ST before bedtime, 2–5 days per week or more was positively associated with increased BP in children aged 5 yo.
Vanderloo LM et al. [25]	2020	Cross-sectional	567	Excessive ST was negatively associated with HDL-c. For every hour of ST, HDL-c decreased by 0.008 mmol/l. Excessive ST was not associated with other individual cardiometabolic risk factors.
Tandon PS et al. [26]	2021	Cross-sectional	1000	Excessive ST was negatively associated with mental health and externalising and internalising symptoms in children and adolescents.
Marin-Dragu S et al. [27]	2023	Cross-sectional	451	Subjective ways of measuring active and passive smartphone use were studied. The metric corresponding to passive use was positively associated with externalising and internalising symptoms. The metric corresponding to active use was negatively associated with externalising symptoms.
Liu M et al. [28]	2022	Meta-analysis	n/a ^1^	Time spent on social media was positively associated with depression. A linear, dose-dependent correlation was established, and it was more prominent in girls.
Hökby S et al. [29]	2023	Longitudinal	4793	Coping mechanisms were studied. Excessive ST was only associated with depression when problem-focused engagement coping was poor. Excessive ST’s association with depression was not moderated by emotion-focused engagement coping.
Zink J et al. [31]	2019	Longitudinal	2525	TV, computer use, and video games were partially bidirectionally linked with emotional disorder symptoms. The relationship was moderated by PA.
Kahn M et al. [32]	2021	Cross-sectional	145	Excessive ST was negatively associated with sleep duration. Excessive ST was positively associated with behavioural problems and externalising symptoms in children aged 3–6 yo only when sleep duration was short.
Tamana SK et al. [33]	2019	Longitudinal	2427	More than 2 h of ST at the age of 3 or 5 was associated with externalising behaviour problems, and attention problems at the age of 5 yo. ST was not associated with internalising behaviour problems.
Lin WH et al. [34]	2020	Longitudinal	2054	Exposure to sexually explicit media at the age of 14 yo was positively associated with early sex debut and unsafe sex at the age of 20 yo.
Lin WH et al. [34]	2020	Longitudinal	1477	Exposure to sexually explicit media at the age of 14 yo was positively associated with multiple sexual partners by the age of 24 yo.
Sajeev MF et al. [37]	2021	Meta-analysis	n/a ^1^	Special VGs were positively associated with less in-hospital paediatric anxiety and pain. They were not associated with less caregiver anxiety.
Rocha LL et al. [38]	2021	Cross-sectional	71,553	More than 2 h of daily ST and eating in front of a TV, computer, or VG was positively associated with ultra-processed food consumption.
Tambalis KD et al. [39]	2020	Cross-sectional	177,091	Excessive ST was positively associated with unhealthy dietary choices, obesity, and adiposity. Excessive ST was negatively associated with sleep duration, PA attributes, and cardiorespiratory fitness.
Helgadóttir B et al. [40]	2021	Cross-sectional	1137	More than 2 h of daily ST was positively associated with skipping breakfast.
Wilksch SM et al. [42]	2019	Cross-sectional	996	Certain social media (differentiated by gender) were associated with thoughts and actions related to eating disorders. Duration of Instagram use was positively associated with eating disorders. Different types of pictures posted on social media were associated differently with eating disorders.
Lonergan AR et al. [43]	2020	Cross-sectional	4209	Certain social media behaviours were positively associated with eating disorders. These behaviours nullified the effect of gender as a moderator.
Kerai S et al. [45]	2022	Cross-sectional	2983	More than 1 h of daily ST was negatively associated with social skills, language and cognitive development, and communication skills in children aged 5 yo.
Madigan S et al. [46]	2020	Meta-analysis	n/a ^1^	Background TV and duration of ST were negatively associated with language skills. Co-viewing, educational ST, and later onset of ST were positively associated with language skills.
Sugiyama M et al. [48]	2023	Longitudinal	885	More than 1 h of ST at the age of 2 y and 8 m was negatively associated with language and daily life skills. The association with daily life skills was mediated by play outside.
Heffler KF et al. [51]	2020	Longitudinal	2152	Passive ST at the age of 12 mo was positively associated with ADHD-like symptoms at the age of 2 yo and not associated with higher ADHD risk at the age of 2 yo. More than 4 h (compared to 3 or less) of passive ST at the age of 18 mo was positively associated with ADHD-like symptoms at the age of 2 yo and not associated with higher ADHD risk at the age of 2 yo.
Moon JH et al. [53]	2018	Cross-sectional	117	Smart-device usage frequency was positively associated with fine motor skills at the age of 3 yo. Appropriate smart-device usage duration was positively associated with social skills at the age of 3 yo. Excessive smart-device usage was negatively associated with expressive (but not total) language skills.
Myers LJ et al. [54]	2016	Longitudinal	60	Video chatting was positively associated with learning and socialising in children aged 2 yo compared to interactive videos. The results were more evident in the middle (17–21 mo) and oldest (aged 22–25 mo) age groups.
Zhu Y et al. [55]	2022	Meta-analysis	n/a ^1^	Parent–child relationship was negatively associated with problematic internet behaviours.
Lin Y et al. [57]	2022	Cross-sectional	827	ST was negatively associated with total sleep time, nighttime sleep, and daytime sleep in infants. It was also negatively associated with total sleep time, nighttime sleep, and daytime sleep in children aged 13–36 mo.
Zhu R et al. [58]	2020	Cross-sectional	2278	More than 1 h of TV ST was positively associated with sleep problems in children aged 3–5 yo. No association was found between computer, iPad, or phone use and sleep problems.
Chang AM et al. ***** [59]	2015	Random crossover clinical trial	12	e-book reading was negatively associated with melatonin levels and REM sleep compared to print-book reading in young adults.
Higuchi S et al. ***** [60]	2005	Random crossover clinical trial	7	VGs before sleep were associated with CNS and ANS arousal.
Exelmans L et al. ***** [61]	2017	Cross-sectional	338	DM displaced sleep in two ways: (i) by delaying bedtime, (ii) by delaying sleeptime
Jain S et al. [62]	2023	Cross-sectional	600	Excessive ST was positively associated with less reading and outdoor play in children. Eye pain was a protective factor against excessive ST.
Dahlgren A et al. [63]	2021	Cross-sectional	121	Objectively measured smartphone use was not associated with PA levels.
Azevedo N et al. [64]	2023	Cross-sectional	1463	Excessive ST was positively associated with back pain in children and adolescents aged 9–19 yo.
Foreman J et al. [65]	2021	Meta-analysis	n/a ^1^	ST was associated with myopia when analysing cross-sectional studies alone (n = 13,431) or cross-sectional and prospective studies combined. ST was not associated when analysing prospective studies alone (n = 3262).
Himebaugh NL et al. ***** [67]	2009	Case control	32	Excessive ST was negatively associated with blinking rate and tear break-up time.
Attygalle UR et al. [68]	2020	Cross-sectional	226	Excessive ST was positively associated with migraine headaches in children. It was not associated with clinically diagnosed ADHD.
Lund J et al. [69]	2022	Cross-sectional	139	Excessive ST was positively associated with migraine headaches with aura in children and adolescents.
Montagni I et al. ***** [70]	2016	Cross-sectional	4927	Excessive ST was positively associated with migraine headaches, and more specifically with migraine headaches without aura in university students.
Baradaran Mahdavi S et al. [71]	2022	Meta-analysis	n/a ^1^	Mobile phone use was positively associated with neck pain in children and adolescents.
Dou Q et al. [72]	2023	Case control	1837	More than 2 h of ST was positively associated with adolescent idiopathic scoliosis in children and adolescents aged 10–18 yo.

^1^ n/a: Not applicable. The criteria were not applied to original studies providing evidence for pathophysiological mechanisms. These studies are marked with an asterisk (“*”).

## Data Availability

Not applicable.

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
