# Peer review of "Screen Time and Its Health Consequences in Children and Adolescents"

_children, 2023, doi:10.3390/children10101665_

Round 1
Reviewer 1 Report
Overall, a comprehensive and detailed analysis. The authors discuss few topics that are in debate regarding the effect of ST on children.
Few suggestions:
1. Please consider adding to the Obesity paragraph the Eating disorders paragraph.
2. The Mental health paragraph is relatively long and contains multiple facts that can be confusing. Please consider focusing on the important or significant research projects.
3. If there is a linkage between any of the other paragraphs, please mention it.
Overall, the article is well written and fluent. One suggestion:
Line 53 - A narrative review was performed - I suggest "Literature review" instead.
Author Response
Dear reviewer,
Thank you very much for your comments. At the end of every paragraph which is related to another one the phrase "Go to the section “name of section” for further relative information." has been added. Additionally, the section of "Mental Health" has been divided to: "3.4.1. Internalising symptoms", "3.4.2. Externalising symptoms", "3.4.3. Suicidal behaviours", and "3.4.4. Healthcare setting". Lastly, minor editing of the English language has been done, while the change in line 53 was addressed.
Addressing the comments from the other reviewer the following changes have been made: The sections "2.2. Inclusion and exclusion criteria", "2.3. Data Extraction" and "3.1. Selection of Studies" have been added in order to describe the selection process of the studies used. A "Discussion" section has also been added. There, the strengths and limitations of the study are presented as well. Moreover, the "Conclusions" section has been moderately modified.
Sincerely,
The authors
Reviewer 2 Report
Dear Author(s),
I have read in detail your paper entitled "Screen Time and its Health Consequences, in Children and Adolescents" submitted to Children. The topic of the paper is extremely important and topical, and there is a lack of high-quality, comprehensive reviews, especially meta-analyses, and systematic reviews.
Although the paper has really great potential I believe that with additional revisions, which I mention below, it could be raised to a higher level of quality and "scientific power".
First, there is no description of how the 69 studies you mention in the abstract were selected. Later in the paper, we see that 47 studies were selected and more meta-analyses, reviews and the like were added. There is a need to further develop and set out the criteria for selecting these studies - I believe that when searching for all these keywords in the databases mentioned, many more papers were displayed than the 69 included. I therefore suggest that the criteria (exclusion and inclusion criteria), the procedure for checking the criteria (e.g., PRISMA Flow) and the like be more clearly defined.
The above is important in the context of the conclusions presented later in the paper. If the papers were chosen "at random", the quality of the results, their generalisability, the conclusions drawn, and the like can be questioned.
Furthermore, after the results are presented, I miss the "discussion" part, where the authors explain the results presented. Since this is a literature review where it is not possible to relate the results to previous research, I consider the above part a valuable part of the paper. In addition, it would be necessary to mention certain limitations of this research that might influence the data presented.
In conclusion, a detailed review of the impact of ST on various aspects of general (bio-psycho-social) functioning was undertaken and the only suggested guideline was to inform parents. Having said the above, I believe that the part of the paper about the implications for future research and the application of the findings in practice can be further explained.
Following all the above, I suggest that the paper be revised and resubmitted for review.
With respect,
Reviewer
Minor editing of English language required
Author Response
Dear reviewer,
Thank you very much for your constructive comments and kind words. The sections "2.2. Inclusion and exclusion criteria", "2.3. Data Extraction" and "3.1. Selection of Studies" have been added in order to better describe the selection process of the studies used. A "Discussion" section has also been added, although most of the findings are presented in the “Results” section. There, the strengths and limitations of the study are documented as well. Moreover, the "Conclusions" section has been moderately modified. Lastly, minor editing of the English language has been done.
Addressing the comments from the other reviewer the following changes have been made: At the end of every paragraph which is related to another one the phrase "Go to the section “name of section” for further relative information." has been added. Additionally, the section of "Mental Health" has been divided to: "3.4.1. Internalising symptoms", "3.4.2. Externalising symptoms", "3.4.3. Suicidal behaviours", and "3.4.4. Healthcare setting"
Sincerely,
The authors
Round 2
Reviewer 2 Report
Dear authors,
I read your revised paper, and it is of much higher quality.
Therefore, I will recommend it for publishing.
Regards,
Reviewer
Author Response
Dear reviewer,
Thank you very much for your positive opinion and kind comments,
Sincerely,
The authors